# Structure of human DPPA3 bound to the UHRF1 PHD finger reveals its functional and structural differences from mouse DPPA3

Nao Shiraishi[1], Tsuyoshi Konuma[2], Yoshie Chiba[3], Sayaka Hokazono[2], Nao Nakamura[1], Md Hadiul Islam[3], Makoto Nakanishi[3], Atsuya Nishiyama [ID][3] & Kyohei Arita [ID][1] ✉

DNA methylation maintenance is essential for cell fate inheritance. In differentiated cells, this involves orchestrated actions of DNMT1 and UHRF1. In mice, the high-affinity binding of DPPA3 to the UHRF1 PHD finger regulates UHRF1 chromatin dissociation and cytosolic localization, which is required for oocyte maturation and early embryo development. However, the human DPPA3 ortholog functions during these stages remain unclear. Here, we report the structural basis for human DPPA3 binding to the UHRF1 PHD finger. The conserved human DPPA3 [85]VRT[87] motif binds to the acidic surface of UHRF1 PHD finger, whereas mouse DPPA3 binding additionally utilizes two unique α-helices. The binding affinity of human DPPA3 for the UHRF1 PHD finger was weaker than that of mouse DPPA3. Consequently, human DPPA3, unlike mouse DPPA3, failed to inhibit UHRF1 chromatin binding and DNA remethylation in *Xenopus egg* extracts effectively. Our data provide novel insights into the distinct function and structure of human DPPA3.

DNA methylation, a cytosine methylation at the 5th carbon atom in a CpG sequence, is a major epigenetic mark that regulates diverse biological processes, including cell-type-specific gene expression, retrotransposon silencing, X-chromosome inactivation, genome imprinting, and carcinogenesis[1,2]. Once DNA methylation patterns are established during cell differentiation, they are faithfully inherited after each replication, to maintain cell identity[3,4]. DNMT1, a maintenance DNA methyltransferase, and ubiquitin-like PHD and RING finger domain-containing protein 1 (UHRF1), also known as Np95/ICBP90, a ubiquitin E3-ligase and recruiter of DNMT1, play pivotal roles in maintaining DNA methylation[5–8]. During this process, the UHRF1 SET- and RING-associated (SRA) domain specifically binds to hemi-methylated DNA[9–11], and UHRF1 ubiquitinates histone H3 or PCNA-associated factor 15 (PAF15) using a plant homeodomain (PHD) finger for recognition, and ubiquitin-like (UBL) and really interesting new gene (RING) domains for multiple mono-ubiquitination[12–17]. Ubiquitinated histone H3 and PAF15 recruit DNMT1 to the late and early replicating domains, respectively[17–19], and stimulate the methyltransferase activity of DNMT1[14,20].

In addition to its well-established role in DNA methylation maintenance, UHRF1 has emerged as a factor in oocyte and preimplantation embryo development[21–23]. A maternal factor, developmental pluripotency-associated 3 (DPPA3), also known as Stella/PGC7, has been identified in mice as a strict inhibitor of chromatin binding of UHRF1 and regulation of its cytosolic localization, in cooperation with exportin-1[24–26]. Expression of mouse DPPA3 (mDPPA3), an intrinsically disordered protein, is restricted to primordial germ cells, oocytes, and preimplantation embryos[24,27,28]. mDPPA3 plays an important role in the formation of oocyte-specific DNA methylation patterns by preventing excessive de novo DNA methylation mediated by UHRF1[24]. Using nuclear magnetic resonance (NMR) solution structural analysis of mouse the UHRF1 PHD finger (mPHD) bound to mDPPA3, we recently revealed that the C-terminal region of mDPPA3 binds to mPHD utilizing a VRT motif at residues 88–90 ([88]VRT[90]), which is conserved in the motifs of other binding partners, histone H3 [1]ART[3] and PAF15 [1]VRT[3] with two subsequent α-helices unique to mDPPA3[29]. Owing to this multifaceted interaction, the binding affinity of mDPPA3 to mPHD ($K_D$ of 0.0277 μM) is significant stronger than those of histone H3 and

[1]Structural Biology Laboratory, Graduate School of Medical Life Science, Yokohama City University, 1-7-29, Suehiro-cho, Tsurumi-ku, Yokohama, Kanagawa 230-0045, Japan. [2]Structural Epigenetics Laboratory, Graduate School of Medical Life Science, Yokohama City University, 1-7-29, Suehiro-cho, Tsurumi-ku, Yokohama, Kanagawa 230-0045, Japan. [3]Division of Cancer Cell Biology, The Institute of Medical Science, The University of Tokyo, 4-6-1 Shirokanedai, Minato-ku, Tokyo 108-8639, Japan. ✉e-mail: aritak@yokohama-cu.ac.jp

PAF15 ($K_D$ of 1.59 μM and 3.52 μM, respectively), indicating that the mechanism by which mDPPA3 inhibits chromatin-binding of UHRF1 involves the competitive binding of between mDPPA3 and histone H3/PAF15 to UHRF1[29]. The biological functions of mDPPA3 as a demethylation factor and UHRF1-inhibitor in oocyte and preimplantation embryos have been extensively studied in mouse models. A recent report has shown that UHRF1 is enriched in the cytoplasmic lattices of human oocytes[30]. However, it is unclear if the biological function of mDPPA3 is conserved in human DPPA3 (hDPPA3), and its role in human oocytes and pre-implantation embryos is unknown. Two α-helices in mDPPA3 which are induced upon binding to mPHD has been shown to be required for the interaction with mUHRF1[29]. However, the amino acid sequences corresponding to these helices are poorly conserved between human and mouse DPPA3 (Fig. 1a), which raises a question of whether hDPPA3 also binds to the hUHRF1 PHD finger in a manner similar to their mouse counterparts, and whether hDPPA3 can inhibit chromatin binding of UHRF1.

In this study, we determined the crystal structure of the human UHRF1 PHD finger complexed with the C-terminal hDPPA3 fragment. The structure clearly showed that the binding mode of hDPPA3 to the human UHRF1 PHD finger differs markedly from that of the mouse proteins and explains why hDPPA3 binds to the human UHRF1 PHD finger with low binding affinity, comparable to the binding of histone H3 and PAF15. Biochemical assays using *Xenopus* egg extracts demonstrated that the inhibitory effect of hDPPA3 on chromatin-binding of UHRF1 is relatively modest compared to the strong inhibition by mouse DPPA3. Our findings shed light on the unexpected role of hDPPA3 in epigenetic regulation during early embryonic development, which differs from the evidence in mice.

## Results

### Interaction between hDPPA3 and hUHRF1 PHD finger

Our previous NMR structural analysis of mDPPA3 complexed with mUHRF1 PHD (mPHD) revealed that residues 85–118 of mDPPA3 are essential for its interaction with mPHD (Figs. 1a and 2b)[29]. Thus, we identified the corresponding region of hDPPA3 by sequence alignment (residues 81–118: hDPPA3$_{81-118}$) (Fig. 1a, b), and evaluated whether this region binds to the human UHRF1 PHD finger, residues 299–366 (hPHD). Isothermal titration calorimetry (ITC) demonstrated that hDPPA3$_{81-118}$ could bind to hPHD with a $K_d$ of 0.868 μM (Fig. 1c and Supplementary Data 1), which is approximately 30-fold weaker than the binding affinity between mDPPA3 and mPHD ($K_d$ = 0.0277 μM)[29]. The binding affinity of hDPPA3$_{81-118}$ to hPHD is comparable with the previously reported binding affinity between hPHD and the histone H3 N-terminal tail (residues 1–15; $K_D$ = 1.7 μM) or PAF15 (residues 1–10; $K_D$ = 2.2 μM)[17,31]. To further investigate the interactions at an atomic resolution, we performed NMR titration experiments. We successfully assigned $^1H$-$^{15}N$ heteronuclear single quantum coherence (HSQC) spectra for [$^{15}N$]-hPHD in the free and complex states with non-labeled hDPPA3$_{81-118}$ (Supplementary Fig. 1a). The $^1H$-$^{15}N$ HSQC spectra of [$^{15}N$]-hPHD titrated with non-labeled hDPPA3$_{81-118}$ showed that the HSQC signals shifted in the intermediate exchange regime on a chemical shift timescale, supporting the modestly weak interaction between hDPPA3$_{81-118}$ and hPHD (Fig. 1d). These data indicate that the binding of hDPPA3 to hPHD was not significantly stronger than that of the other binding partners, histone H3 and PAF15. Chemical shift differences (CSD) between the free and complexed states showed relatively large values for Asp330, Met332, Asp337, Glu355, and Asp356, suggesting the contribution of the main chain of these amino acid residues to the hPHD–hDPPA3 interaction (Supplementary Fig. 1b).

### Crystal structure of hDPPA3 bound to hPHD

To reveal the molecular basis for the binding mode of hDPPA3 to hPHD, we determined the crystal structure of hDPPA3$_{81-118}$ in complex with hPHD at a 2.4 Å resolution (Table 1). The asymmetric unit contained one hPHD:hDPPA3$_{81-118}$ complex, and a $2|F_o| - |F_c|$ map corresponding to residues 299–363 of hPHD and residues 84–107 of hDPPA3$_{81-118}$ was unambiguously observed (Supplementary Fig. 2a). hPHD consists of pre-

and core-PHD domains that include three zinc finger motifs (Fig. 2a)[31]. The structure of the hPHD moiety in the complex with hDPPA3$_{81-118}$ was well-superimposed on apo-hPHD (PDB:3SOX, root mean square deviation [RMSD] of Cα atoms with 0.848 Å) and those in the complex with histone H3 (PDB:3ASL, RMSD: 0.795 Å) and PAF15 (PDB:6IIW, RMSD: 0.397 Å), implying that the binding of hDPPA3 does not undergo conformational changes in hPHD (Supplementary Fig. 2b).

In contrast to the hPHD moiety, the binding mode of hDPPA3 shares both similarities and dissimilarities with that of mDPPA3 (Fig. 2a, b). The conserved VRT motif at residues 85–87 of hDPPA3$_{81-118}$ is accommodated on the acidic surface of hPHD, the binding site for $^1ART^3$ of histone H3, and $^1VRT^3$ of PAF15, in a manner concordant with the motif in mDPPA3 ($^{88}VRT^{90}$) (Fig. 2b). The side chain of Val85 in hDPPA3$_{81-118}$ forms a hydrophobic interaction with Leu331, Val352, Pro353, and Trp358 in hPHD (Fig. 2a). The positively charged guanidino group at Arg86 of hDPPA3$_{81-118}$ forms hydrogen bonds with the side chains of Asp334 and Asp337 of hPHD (Fig. 2a, c). The side chain methyl and hydroxyl groups of Thr87 in hDPPA3$_{81-118}$ forms hydrophobic interactions with Leu331 and Val352 of hPHD and hydrogen bonds with the main chain amide of Ser90 of hDPPA3$_{81-118}$ (Fig. 2a). The latter potentially functions as a helical cap for the N-terminus of the following α-helix (Fig. 2a). Leu88 of hDPPA3$_{81-118}$ is surrounded by the side chains of Ala317, Gln330, Met332, and Ala339 in hPHD, in which the side chain of Met332 functions as a separation between the side chains of Arg86 and Leu88 of hDPPA3$_{81-118}$ (Fig. 2a).

When mPHD binds to mDPPA3, the two α-helices following the VRT motif of mDPPA3 form an L-like shape, in which the long α-helix binds to the shallow groove between the pre- and core-PHD fingers (Fig. 2b). However, the C-terminus of the $^{85}VRT^{87}$ motif of hDPPA3$_{81-118}$ forms a unique conformation that differed from that of mDPPA3. Residues 88–101 of hDPPA3 forms a four-turn single α-helix, which is not kinked and markedly differs from mDPPA3 complexed with mPHD (Fig. 2a, b). The contact area between the hPHD and hDPPA3 (ca. 449 Å²) was smaller than that of the mouse protein (ca. 1360 Å²)[32], which is concordant with the weaker dissociation constant of the human proteins.

### Structural feature of hPHD:hDPPA3 in solution

Intriguingly, the α-helix of hDPPA3$_{81-118}$ has no contact with the hPHD moiety in the crystal (Fig. 2a, c). Instead, the α-helix interacts with the corresponding part of a symmetry molecule related to a crystallographic two-fold axis (Supplementary Fig. 2c). This interaction in the crystal gives rise to two possibilities: the helical structure formation of hDPPA3 is an artifact of crystal packing, or the hPHD:hDPPA3 complex forms a dimer structure via the interaction mediating the α-helix of hDPPA3.

Next, we examined the structure of hDPPA3$_{81-118}$ in solution using circular dichroism (CD) and size-exclusion chromatography in line with small angle X-ray scattering (SEC-SAXS) which can analyze the solution structure, oligomeric state, conformational changes and flexibility of bio-macromolecules at a scale ranging from a few Å to hundreds of nm (Supplementary Fig. 3a–c and Supplementary Table 1)[33]. The CD spectrum exhibited that hDPPA3$_{81-118}$ alone showed a typical random-coil spectrum (Fig. 3a and Supplementary Data 2). The CD spectrum of hDPPA3$_{81-118}$ mixed with hPHD showed a negative peak at 222 nm, which was lower than the sum of the spectra of hPHD and hDPPA3$_{81-118}$ alone (Fig. 3a and Supplementary Data 2), indicating that the binding of hDPPA3 to hPHD involved a coupled folding and binding mechanism. The SEC-SAXS data also supported the coupled folding and binding mode of hDPPA3. The dimensionless Kratky plot showed the unfolding state of sole hDPPA3$_{81-118}$, whereas the hPHD:hDPPA3$_{81-118}$ complex was in a globular state (Fig. 3b and Supplementary Data 3).

SEC-SAXS experiments also revealed that the molecular mass of the measured proteins was estimated by the empirical volume of correlation $Vc$[34], resulting in a 13.0 kDa hPHD:hDDPA3$_{81-118}$ complex, which was highly similar to the molecular weight calculated from the amino acid sequence of the hPHD:hDPPA3$_{81-118}$ complex with 1:1 stoichiometry (12.2 kDa) (Supplementary Table 1). The ab initio model of the measured

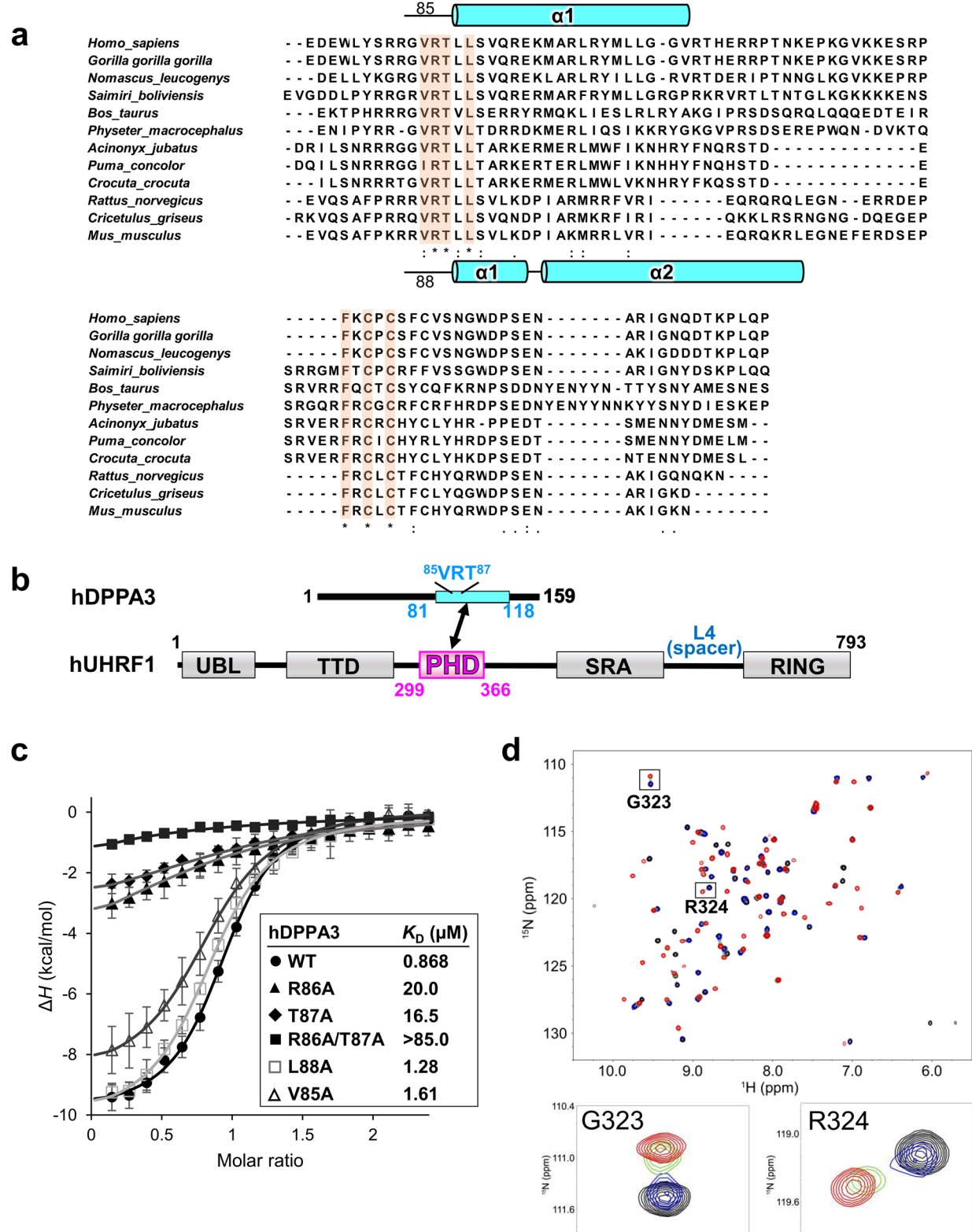

**Fig. 1 | Characterization of the interaction between hUHRF1 and hDPPA3.**
**a** Amino acid sequence alignment of C-terminal part of DPPA3. Secondary structures of mouse and human DPPA3 are indicated based on PDB:7XGA and analysis of this study, respectively. **b** Schematic of the domain composition of human UHRF1 and DPPA3. **c** Isothermal titration calorimetry measurements for hPHD and wild-type (WT)/mutants of hDPPA3$_{81-118}$. Superimposition of enthalpy change plots with standard deviations. Data were presented as mean values for $n = 3$. **d** Overlay of $^1$H-$^{15}$N heteronuclear single quantum coherence (HSQC) spectra of 30 μM hPHD showing chemical shift changes upon titration with hDPPA3$_{81-118}$ of 0 μM (black), 15 μM (blue), 30 μM (green), and 60 μM (red). Square regions inside the HSQC spectra were expanded (lower panels).

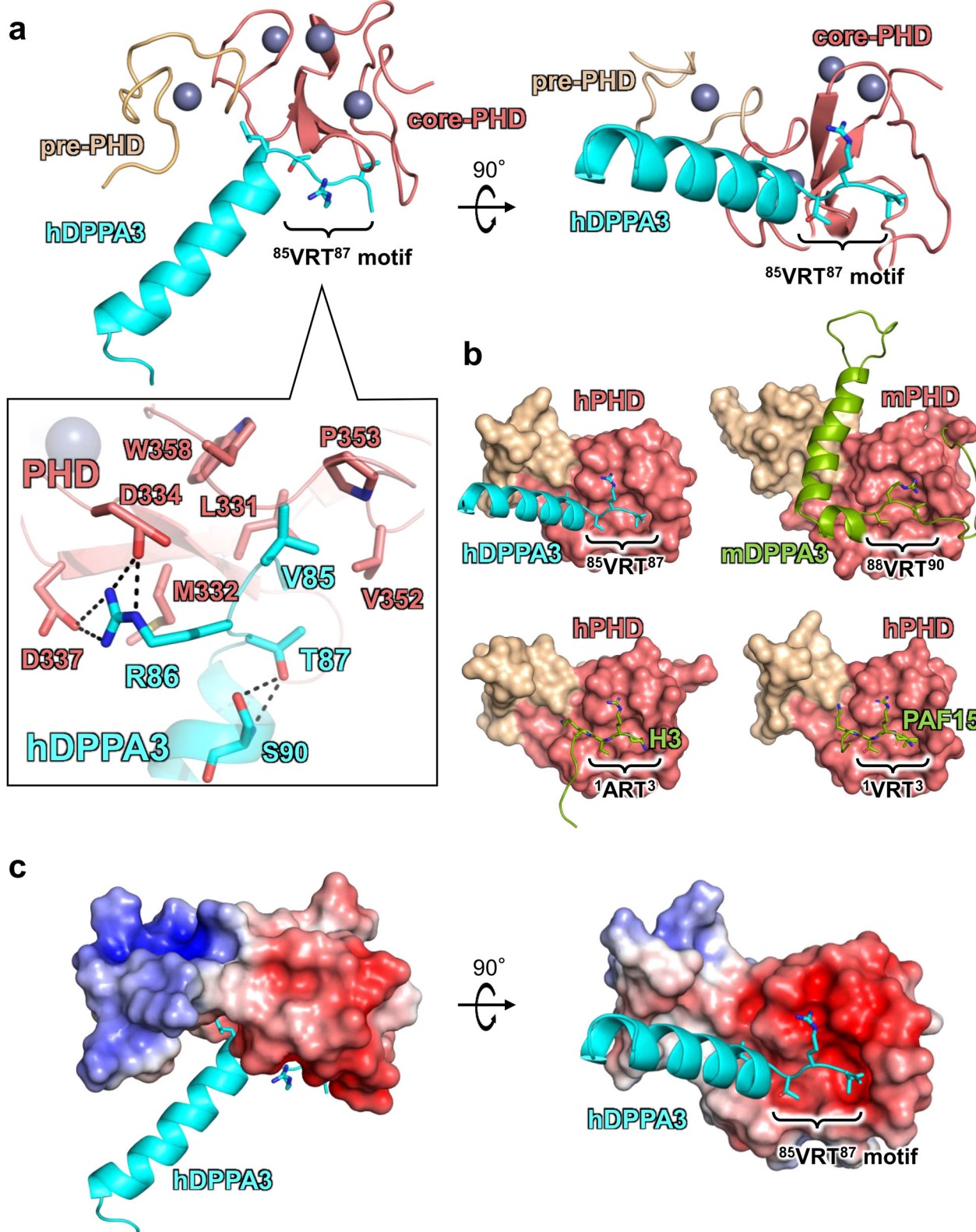

**Fig. 2 | Crystal structure of hDPPA3$_{81-118}$ in complex with hPHD. a** Overall structure of hPHD:hDPPA3$_{81-118}$ complex. Pre-PHD, core-PHD and hDPPA3 are depicted as gold, salmon, and cyan cartoon models, respectively. The conserved VRT motif in hDPPA3 is displayed as a stick model. Inset shows the interaction between the VRT motif of hDPPA3 and hPHD. The black dotted line represents a hydrogen bond. **b** Structural comparison of hPHD:hDPPA3 (this study, upper left),

mPHD:mDPPA3 (PDB: 7XGA, upper right), hPHD:H3 (PDB: 3ASL, bottom left) and hPHD:PAF15 (PDB: 6IIW, bottom right) complexes. mDPPA3, H3 and PAF15 are shown as a green cartoon model and VRT (ART) motif are represented as stick model. **c** Electrostatic surface potential of hPHD calculated with program APBS[56]. The red and blue surface colors represent negative and positive charges, respectively. hDPPA3 is depicted as a cyan cartoon.

## Table 1 | Data collection and refinement statistics

|  | hPHD:hDPPA3 (PDB:8WMS) |
|---|---|
| **Data collection** |  |
| Space group | $I4_122$ |
| **Cell dimensions** |  |
| $a, b, c$ (Å) | 77.80 77.80 140.67 |
| Resolution (Å) | 43.33–2.40 (2.49–2.40)[a] |
| $R_{meas}$ (%) | 7.0 (53.7)[a] |
| $R_{pim}$ (%) | 2.6 (21.0)[a] |
| Mean ($I/\sigma(I)$) | 12.5 (2.5)[a] |
| $CC_{1/2}$ | 99.9 (89.6)[a] |
| Completeness (%) | 99.7 (99.6)[a] |
| Redundancy | 5.4 (5.7)[a] |
| Total reflections | 47,813 (5172)[a] |
| Unique reflections | 8775 (903)[a] |
| **Refinement** |  |
| Resolution (Å) | 43.33–2.40 |
| No. reflections | 8693 |
| $R_{work}/R_{free}$ (%) | 23.3/26.6 |
| **No. atoms** |  |
| hPHD | 514 |
| hDPPA3 | 190 |
| zinc | 4 |
| **$B$ factors (Å$^2$)** |  |
| hPHD | 79.9 |
| hDPPA3 | 83.5 |
| zinc | 67.6 |
| **R.m.s. deviations** |  |
| Bond lengths (Å) | 0.004 |
| Bond angles (°) | 0.803 |

[a]() Values in parentheses are for the highest-resolution shell.

proteins showed clear results. The overall shape of the bead model was well superimposed on the crystal structure of the hPHD:hDPPA3$_{81-118}$ complex in the asymmetric unit (Fig. 3c, d and Supplementary Data 3). These data indicated that hDPPA3 binds to hPHD at 1:1 stoichiometry with the induction of a four-turn single α-helix.

### Validation of the structural data by mutational analysis

To validate our structural data and confirm the contribution of individual residues to complex formation, ITC experiments were conducted using hPHD and hDPPA3 harboring mutations in the VRT sequence. Mutations with deleterious effects on the interaction were R86A and T87A of hDPPA3, which reduced the dissociation constant to 20.0 and 16.5 μM, respectively, and a double mutation (R86A/T87A), which resulted in a more severe reduction in the interaction, with a $K_D$ exceeding 85.0 μM (Fig. 1c and Supplementary Data 1). In contrast, alanine mutations at Val85 and Leu88 in hDPPA3 had less marked effects on the hPHD:hDPPA3 interaction (Fig. 1c and Supplementary Data 1).

We further investigated mutants of DPPA3 that affect dimer formation as observed in the crystal structure (Supplementary Fig. 4a and Supplementary Data 1). R98A/M102A, located at C-terminal region in the α-helix of hDPPA3 and potentially interacting with hPHD of the symmetrical molecule in the crystal, did not reduced the binding affinity. Similarly, M96A/L99A, which contribute to the formation of the helix bundle of hDPPA3 in the crystal, also had no effect on the interaction with hPHD, validating the 1:1 stoichiometry of the hPHD:hDPPA3 complex in solution. Interestingly, the introduction of proline residue, known as a helix breaker,

at both Arg93 and Ala97 in hDPPA3 (R93P/A97P) significantly reduced the binding affinity to hPHD, with $K_D$ of 9.39 μM (Supplementary Fig. 4a and Supplementary Data 1), indicating that helical structural formation following the VRT motif in hDPPA3 is crucial for its interaction with hPHD.

Next, mutations were introduced into hPHD. Concordant with the hDPPA3 mutants, the D334A/D337A mutations in hPHD, which form an ionic-pair with Arg86 of hDPPA3, had a severe effect, reducing the binding affinity to a $K_D$ exceeding 115 μM. The M332A hPHD mutation showed a decreased binding affinity, with a $K_D$ of 8.07 μM (Supplementary Fig. 4a and Supplementary Data 1). ITC data based on mutant proteins indicate that the VRT motif of hDPPA3 is important for its interaction with the UHRF1 hPHD finger.

### Effect of hDPPA3 on UHRF1 function

Next, to analyze whether hDPPA3 affects the biological functions of UHRF1, ubiquitination of histone H3, and chromatin binding, we performed NMR titration assays and in vitro biochemical experiments using recombinant proteins and *Xenopus* egg extracts. First, we examined the competitive binding of hDPPA3 and histone H3 to hPHD because both hDPPA3 and histone H3 mainly bind to the acidic surface of hPHD via the $^{85}$VRT$^{87}$ and $^{1}$ART$^{3}$ motifs, respectively. We conducted NMR titration experiments using $^{1}$H-$^{15}$N labeled hPHD and non-labeled hDPPA3$_{81-118}$ and/or histone H3 peptides (residues 1–37W; the H3$_{1-37W}$ peptide). The HSQC spectrum of hPHD mixed with hDPPA3$_{81-118}$ and the H3$_{1-37W}$ peptide (hPHD:hDPPA3:H3 = 1:2:2) showed most of the signals, with weakened or no intensity by the broadening due to chemical exchange, suggesting that, as expected, hDPPA3$_{81-118}$ and the H3$_{1-37W}$ peptide competitively bound to the acidic surface of hPHD as the shared binding site (Fig. 4a, upper). In the presence of excess H3$_{1-37W}$ peptide (hPHD:hDPPA3$_{81-118}$:H3 = 1:2:8), hDPPA3$_{81-118}$ could not bind to hPHD (Fig. 4a, lower). This differed from the situation with mDPPA3, which bound to mPHD even in the presence of excess H3$_{1-37W}$ peptide (mPHD:mDPPA3:H3 = 1:2:8)[29]. An in vitro ubiquitination assay of C-terminal FLAG-tagged H3$_{1-37W}$ with full-length human UHRF1 also supported the weak inhibitory effect of hDPPA3. hDPPA3 did not effectively inhibit ubiquitination of the histone H3 tail, whereas mDPPA3 showed a markedly negative effect on ubiquitination (Supplementary Fig. 4b and Supplementary Data 4). The addition of hDPPA3 to a 1–2 equimolar excess of histone H3 only slightly inhibited histone H3 ubiquitination (Fig. 4b and Supplementary Data 4). Mutant forms of hDPPA3, which exhibited decreased binding to hPHD, failed to inhibit ubiquitination of histone H3 (Fig. 4b and Supplementary Data 4). These findings indicate that the binding of hDPPA3$_{81-118}$ to UHRF1 inhibits the ubiquitination activity of UHRF1 on histone H3; however, the inhibitory effect was moderately weak due to the low binding affinity between hDPPA3$_{81-118}$ and UHRF1.

Finally, we tested the ability of hDPPA3 to inhibit UHRF1 chromatin binding in *Xenopus* egg extracts (Fig. 5a). As previously reported, the addition of 0.5 μM recombinant mDPPA3 to interphase extracts was sufficient to block UHRF1 chromatin loading, UHRF1-dependent PAF15 ubiquitylation, and DNMT1 recruitment (Fig. 5b and Supplementary Data 4). In contrast, hDPPA3 did not inhibit the chromatin binding of UHRF1 and DNMT1 recruitment, even at 1.0 μM (Fig. 5b and Supplementary Data 4). Consistently, hDPPA3 did not show significant inhibitory activity on DNA methylation in *Xenopus* egg extracts compared to mDPPA3 (Fig. 5c).

Taken together, the binding of hDPPA3 to hUHRF1 PHD competes with that of histone H3. However, it is noteworthy that the inhibitory effect exerted by hDPPA3 was relatively modest, implying that hDPPA3 does not appear to function as a strong inhibitor of UHRF1 chromatin binding, unlike mouse DPPA3.

### Discussion

Our structural analysis revealed that human DPPA3 binds to hPHD solely through a conserved VRT sequence motif. This finding is consistent with

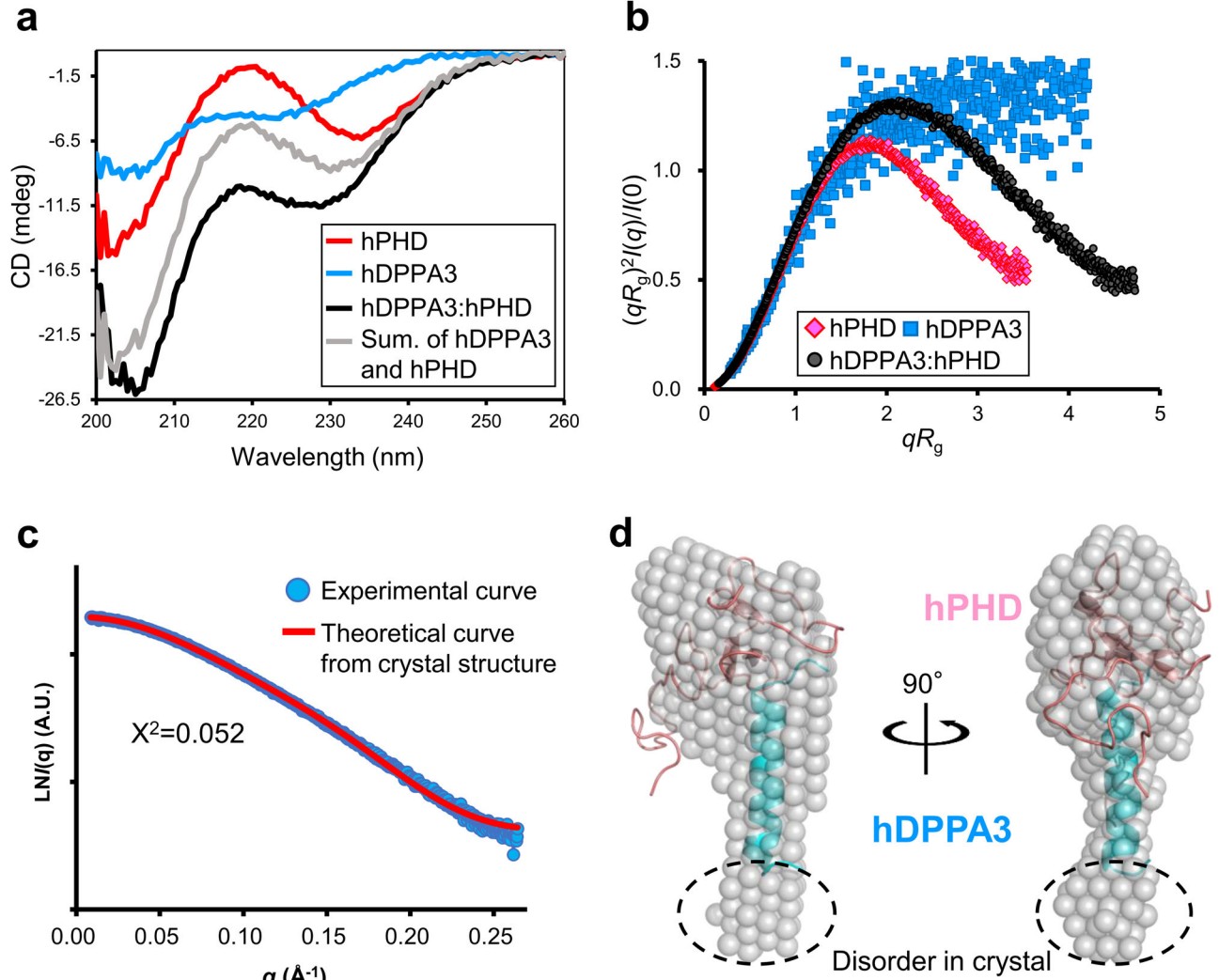

**Fig. 3 | Solution structure of hDPPA3. a** CD spectra of hPHD alone (red), hDPPA3$_{81-118}$ alone (blue), and the hPHD in complex with hDPPA3$_{81-118}$ (black). The sum of CD spectra of hPHD alone and hPDDA3$_{81-118}$ alone is shown as gray. **b** Dimensionless Kratky plots of hPHD alone (red diamond), hDPPA3$_{81-118}$ alone (blue square), and hPHD in complex with hDPPA3$_{81-118}$ (black circle) derived from small-angle X-ray scattering (SAXS) data. **c** Comparison of scattering curve derived

from experimental data (cyan) and theoretical curve of the crystal structure of the hPHD:hDPPA3$_{81-118}$ complex (red). **d** Structural comparison of solution and crystal structures of the hPHD:hDPPA3$_{81-118}$ complex. Ab initio bead model of the hPHD:hDPPA3$_{81-118}$ complex derived from the SAXS scattering data (transparent gray sphere) is superimposed on the crystal structure (cartoon).

biochemical data showing that the binding affinity of hDPPA3 to hPHD was in the sub-micro order range of $K_D$, with an approximately 30-fold decrease in the binding affinity of its mouse protein counterpart. The weak binding affinity of hDPPA3 was insufficient to inhibit the chromatin binding of UHRF1 in *Xenopus* egg extracts. Our data suggested that the inhibitory effect of hDPPA3 differs from that of mDPPA3 under similar conditions. This raises the question of whether hDPPA3 can act as an inhibitor of UHRF1 in human oocytes and early embryogenesis. There are several possibilities to consider in this regard. Intrinsically disordered protein (region) containing low complexity sequence frequently associates with formation of liquid-liquid phase separation (LLPS)[35]. Notably, sequence analysis of human and mouse DPPA3 using FuzDrop (https://fuzdrop.bio. unipd.it) indicated that human DPPA3 exhibits a higher potential for droplet formation that mouse DPPA3 (Supplementary Fig. 5)[36]. This prediction suggests that condensed hDPPA3 within the droplet may preferentially bind to UHRF1, thereby inhibiting the chromatin binding of UHRF1. In another situation, the level of hDPPA3 protein expression in human oocytes and zygotes is key to the inhibition of UHRF1 chromatin binding. Our data indicated that the binding affinity of hDPPA3 for hPHD was approximately 1.7-fold stronger than that of histone H3, suggesting that

a locally high concentration of hDPPA3 contributes to its preferential binding to hUHRF1 to inhibit chromatin binding. Another possibility involves post-translational modifications of histone H3. Given that the methylation of Arg2 and phosphorylation of Thr3 in histone H3 greatly impair its binding to the UHRF1 PHD finger[31], hDPPA3 might bind to UHRF1 even at low protein concentrations. Recently, NLRP14 (Nucleotide-binding oligomerization domain, leucine-rich Repeat and Pyrin domain containing) has emerged as a factor related to reproduction. It interacts with UHRF1 in the zygote and two-cell stages in the cytosol[21,37]. If cytoplasmic localization of UHRF1 is not mediated by hDPPA3, it may be important for UHRF1 to interact with NLRP14 immediately after its translation into the cytoplasm. Interestingly, the cytosolic localization of the mRNA of a guanine nucleotide exchange factor, NET1, has been reported to regulate protein–protein interactions after translation, ultimately determining protein localization[38].

The VRT motif in DPPA3, which binds to the acidic surface of the UHRF1 PHD finger, is well conserved across various species (Fig. 1a). Conversely, the amino acid sequence corresponding to the α-helix following the VRT motif showed significant diversity. Interestingly, AlphaFold2 (AF2) structural prediction indicated that mDPPA3 has both short and long α-

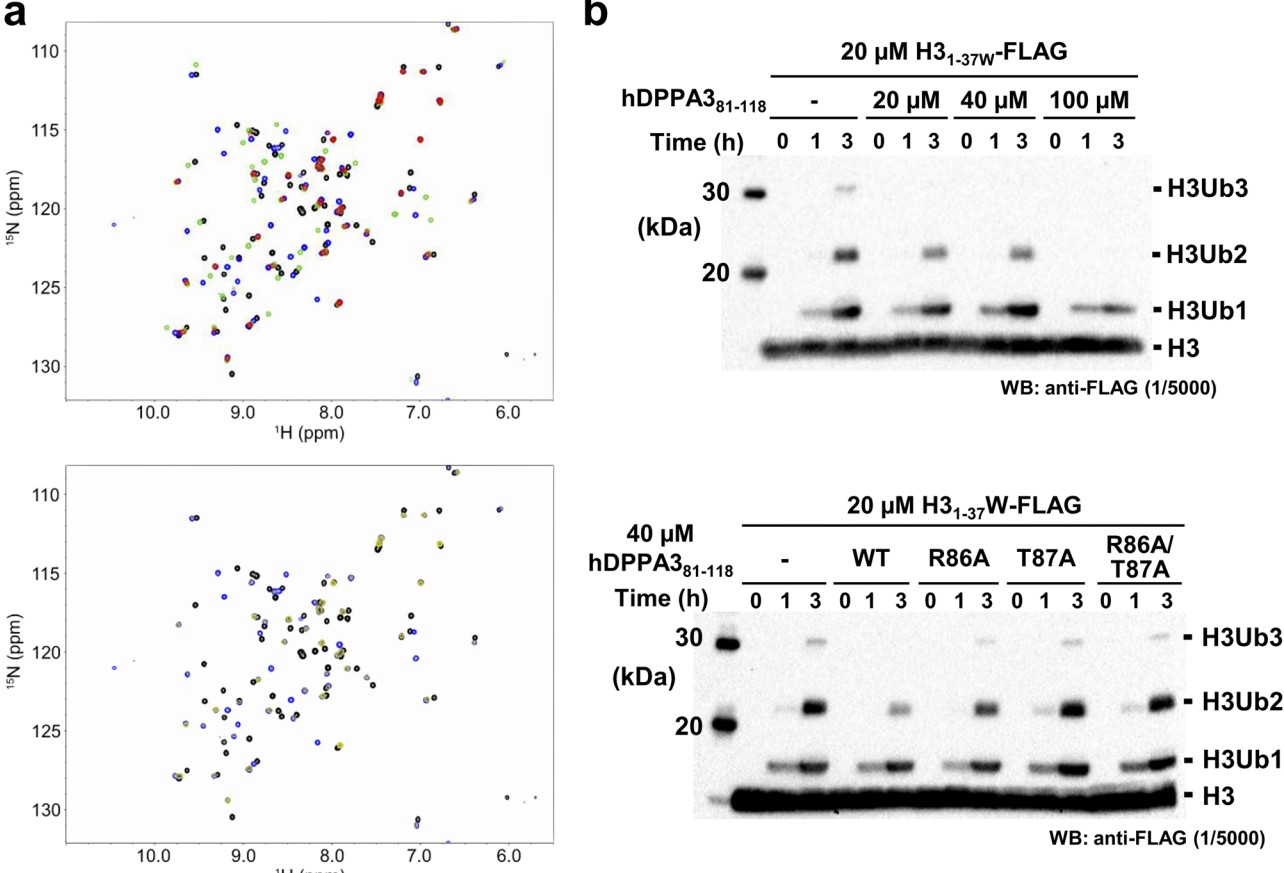

**Fig. 4 | Competitive assay between hDPPA3 and the histone H3 tail. a** Overlay of $^1H$-$^{15}N$ HSQC spectra of $^{15}N$-labeled hPHD in the presence of hDPPA3$_{81-118}$ and/or the H3$_{1-37W}$ peptide at a molar ratio of 1:0:0 (black), 1:2:0 (green), 1:0:2 (blue), and 1:2:2 (red) of hPHD:hDPPA3:H3 (upper), and of 1:0:0 (black), 1:0:2 (blue), and 1:2:8 (yellow) of hPHD:hDPPA3:H3 (lower). **b** In vitro ubiquitination assay. C-terminal FLAG tagged-H3$_{1-37W}$ was ubiquitinated using in-house purified E1, E2, and human UHRF1 (E3). The ubiquitinated H3 was detected using anti-FLAG antibody. Upper panel shows that 20, 40, and 100 μM hDPPA3$_{81-118}$ was added to the reaction solution including 20 μM of H3$_{1-37W}$. The lower panel presents results of an in vitro ubiquitination assay using 40 μM hDPPA3$_{81-118}$ mutants. The gel image is representative of $n = 3$ independent experiments.

helices following the VRT motif, forming an L-like shape, consistent with our NMR structure of the mPHD:mDPPA3 complex (Supplementary Fig. 6)[39]. In contrast, human DPPA3 exhibits a single long α-helix at the same position. AF2 predictions also suggest that *Homo sapiense* (UniProt ID: Q6W0C5), *Bos taurus* (A9Q1J7), *Gorilla gorilla gorilla* (G3RB81), *Saimiri boliviensis* (A0A2K6SNG1), *Puma concolor* (A0A6P6HCW6), *Nomascus leucogenys* (A0A2I3H008), *Crocuta crocuta* (A0A6G1B388), *Physeter macrocephalus* (A0A2Y9EH83), and *Acinonyx jubatus* (A0A6I9ZFC3) possess a single α-helix, while *Mus musculus* (Q8QZY3), *Rattus norvegicus* (Q6IMK0), and *Cricetulus griseus* (A0A3L7H856) have two α-helices, consisting of both short and long α-helices, as far as we could find in the database (Supplementary Fig. 6). These observations suggest the potential limitation of the two α-helices to Rodentia and underscore the utility of AF2 structural prediction for the classification of the DPPA3 function based on the helical content. The major difference in the helical region of human and mouse DPPA3 is the substitution of a proline residue with a lysine residue at the 95th position of human DPPA3 (Fig. 1a). A similar substitution is also found in the species that predictably forms as single α-helix. However, the K95P mutation in human DPPA3 did not enhance its binding affinity for hPHD (Supplementary Fig. 7a). AF2 prediction of the K95P mutant of hDPPA3-suggested that a single α-helix remains the predominant conformation (Supplementary Fig. 7b), suggesting that the differences in the helical structural regions of human and mouse DPPA3 are governed by more complicated mechanisms than a simple amino acid substitution.

The distinctive α-helical arrangement in hDPPA3 revealed in our structural analysis shed light on the function of this protein in oocytes and preimplantation embryo development distinct from the mouse DPPA3. Our results encourage further investigations into the functional implications of hDPPA3, potentially paving the way for novel discoveries in this context.

## Methods

### Peptides and primers

The human DPPA3 peptide, residues 81–118 (NH$_2$-$^{81}$SRRGVRTLLSVQREKMARLRYMLLGGVRTHERRPTNKE$^{118}$-COOH) for crystallography and K95P substituted hDPPA3$_{81-118}$ for ITC experiment were purchased from Toray Research Center (Tokyo, Japan). Primers for site-directed mutagenesis of hDPPA3 are listed as follows:

V85A (Forward: 5′-GAGAGGAGCAAGAACATTGCTGTCTGTGCA-3′, Reverse: 5′-ATGTTCTTGCTCCTCTCCTGCTCCCACCTC-3′),

R86A (Forward: 5′- AGGAGTAGCAACATTGCTGTCTGTGCAGAG-3′, Reverse: 5′- GCAATGTTGCTACTCCTCTCCTGCTCCCAC-3′),

T87A (Forward: 5′- AGTAAGAGCATTGCTGTCTGTGCAGAGAGA-3′, Reverse: 5′- ACAGCAATGCTCTTACTCCTCTCCTGCTCC-3′),

L88A (Forward: 5′- AAGAACAGCGCTGTCTGTGCAGAGAGAAAA-3′, Reverse: 5′- CAGACAGCGCTGTTCTTACTCCTCTCCTGC-3′)

R86A/T87A (Forward: 5′- AGGAGTAGCAGCATTGCTGTCTGTGCAGAG-3′, Reverse: 5′- ACAGCAATGCTGCTACTCCTCTCCTGCTCC-3′).

R98A/M102A (Forward: 5′- GGCAGCATTGAGATACGCGTTACTCGGCGGAGTTC -3′, Reverse: 5′- GTAACGCGTATCTCAATGCTGCCATCTTTTCTCTC -3′).

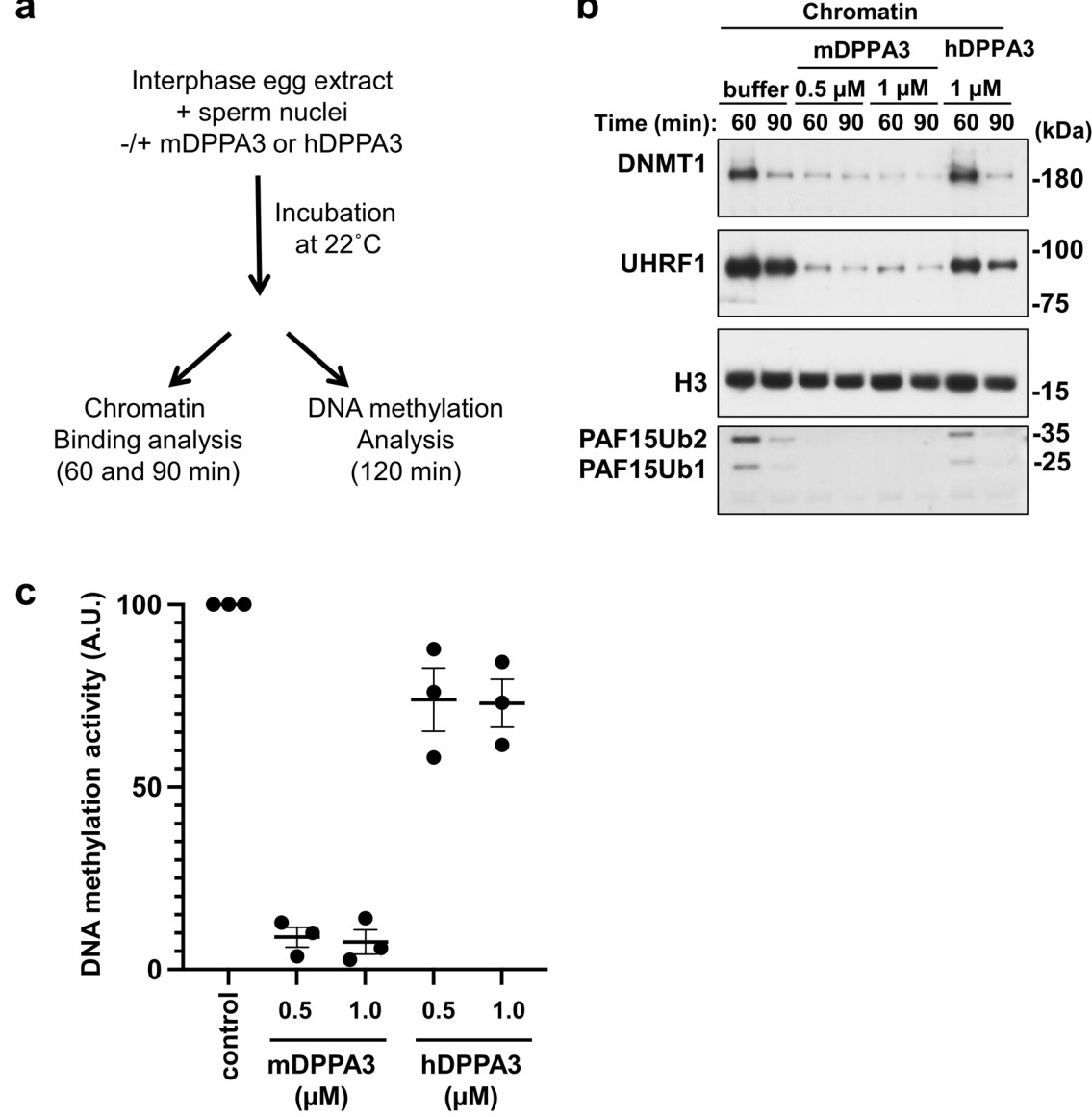

**Fig. 5 | Functional assay of DPPA3 using *Xenopus* egg extracts. a** Experimental design for functional analysis of DPPA3 using *Xenopus* egg extracts. **b** Sperm chromatin was incubated with interphase *Xenopus* egg extracts supplemented with buffer (+buffer), 3×FLAG-mDPPA3, or 3×FLAG-hDPPA3. Chromatin fractions were isolated and immunoblotted using the indicated antibodies. The gel image is representative of $n = 3$ independent experiments. **c** Sperm chromatin was added to interphase egg extracts supplemented with radiolabeled S-[methyl-$^3$H]-adenosyl-L-methionine and buffer (control), 3×FLAG-mDPPA3, or 3×FLAG-hDPPA3. The efficiency of DNA methylation maintenance was assessed by the incorporation of radio-labeled methyl groups from S-[methyl-$^3$H]-adenosyl-L-methionine ($^3$H-SAM) into DNA purified from the egg extracts. Data were presented as mean values ± SD for $n = 3$.

M96A/L99A (Forward: 5′- AAAGGCGGCAAGAGCGAGATACAT GTTACTCGGCG -3′, Reverse: 5′- ATCTCGCTCTTGCCGCCTTTTCT CTCTGCACAGAC -3′).

R93P/A97P (Forward: 5′- GCAGCCAGAAAAGATGCCAAGATT GAGATACATGT -3′, Reverse: 5′- ATCTTGGCATCTTTTCTGGCTGC ACAGACAGCAAT -3′).

**Protein expression and purification**
Human UHRF1 PHD finger (residues 299–366) for crystallography, SAXS, NMR, CD, and ITC experiments was expressed in *Escherichia coli* (*E.coli*) and purified according to previous paper[31]. Briefly, hPHD was expressed as a GST-fusion protein and purified using glutathione Sepharose 4B (GS4B), anion exchange (HiTrap Q) and 26/600 Superdex 75 chromatography (Cytiva). hDPPA3, residues 81–118, for SAXS, NMR, CD, ITC and ubiquitination experiments was expressed as a six histidine-tagged ubiquitin (His-Ub) fusion protein. The protein was expressed in *E. coli* BL21 (DE3) in

Luria–Bertani medium (LB) containing 12.5 µg/ml kanamycin. When the optical density at 660 nm (O.D.660) of the cells reached 0.7, 0.4 mM iso-propyl β-d-thiogalactoside (IPTG) was added to the medium and the cells were further harvested for 6 h at 30 °C. The cells were suspended in lysis buffer (40 mM Tris-HCl [pH7.5], 400 mM NaCl and 30 mM imidazole). After cell lysis by sonication and removal of cell debris by centrifugation, the supernatant was loaded onto a histidine-tag affinity column Ni Sepharose 6 Fast Flow (Cytiva), and the sample was eluted from the column using elution buffer containing 500 mM imidazole. Next, the His-Ub tag was removed by Saccharomyces cerevisiae ubiquitin carboxyl-terminal hydrolase YUH1. The sample was further purified using HiTrap SP HP cation-exchange chromatography (Cytiva) and finally purified using HiLoad 26/600 Superdex 75 size-exclusion chromatography equilibrated with 1 × ITC buffer (10 mM HEPES (pH7.5), 150 mM NaCl, 0.25 mM tris (2-carbox-yethyl)phosphine (TCEP)). The H3 peptide (residues 1–36 with an additional tryptophan residue at their C-terminus, hereafter H3$_{1-37W}$), mouse

DPPA3, full-length mouse UHRF1, full-length human UHRF1, mouse UBA1 and human UBE2D3 for the in vitro ubiquitination assay were purified according to previous reports[17,29].

For the preparation of $^{15}$N-labeled or $^{13}$C,$^{15}$N-double labeled hPHD, M9 minimal media containing 0.5 g/l $^{15}$NH$_4$Cl or 0.5 g/l $^{15}$NH$_4$Cl and 1 g/l $^{13}$C-glucose was used instead of LB media. Site-directed mutagenesis of hPHD and hDPPA3$_{81-118}$ was performed by designing two primers containing the mutations. The mutants of hDPPA3$_{81-118}$ and the labeled hPHD were purified using the same protocol. The mutants of hDPPA3$_{81-118}$ and the labeled hPHD were purified by the same protocol.

## Crystallography of hPHD in complex with hDPPA3 peptide

The hPHD:hDPPA3$_{81-118}$ complex was prepared by adding an equi-molar excess of hDPPA3 peptide to hPHD prior to crystallization. The crystal was obtained using an 8 mg/ml concentration of the complex at 4 °C and the hanging drop vapor diffusion method with a reservoir solution containing 100 mM Tris-HCl (pH 8.5) and 2 M Ammonium sulfate. The crystal was directly frozen in liquid nitrogen using a cryoprotectant containing 25% (v/v) ethylene glycol. The X-ray diffraction data were collected at a wavelength of 0.98000 Å on a Pilatus3 6 M detector in beam line BL-17A at Photon Factory (Tsukuba, Japan) and scaled at 2.40 Å resolution using the program XDS package[40] and Aimless[41]. After molecular replacement by PHASER[42] using human PHD finger (PDB: 3ASL) as a search model and several cycles of model refinement by PHENIX[43], the final model converged at 2.40 Å resolution with a crystallographic R-factor of 23.3% and a free R-factor of 26.6%.

The crystallographic data and refinement statistics are given in Table 1. Figures were generated using PyMol (http://www.pymol.org).

## NMR

All NMR experiments were performed using a Bruker BioSpin Avance III HD spectrometers with TCI triple-resonance cryogenic probe-heads and basic $^1$H resonance frequency of 600.03 and 800.23 MHz. Three-dimensional (3D) spectra for backbone signal assignments, including HNCACB, CACB(CO)NH, HNCA, HN(CO)CA, HNCO, and HN(CA)CO, were acquired at 293 K for 520 μM [$^{13}$C, $^{15}$N]-hPHD dissolved in PBS buffer (pH 7.0) containing 1 mM DTT and 5% D$_2$O. For the complex state, 260 μM [$^{13}$C, $^{15}$N]-hPHD with hDPPA3$_{811-118}$ at molar ratio of 1:2 was used in the buffer same as the free state. The spectral widths (total number of data points) of each spectrum were 18 ppm (2048) for the $^1$H dimension and 24 ppm (192) for the $^{15}$N dimension. All 3D spectra were acquired using non-uniform sampling (NUS) to randomly reduce the t$_1$ and t$_2$ time-domain data points by 25%. The uniformly sampled data were reconstructed from the raw NMR data using various techniques such as IST or SMILE[44,45]. All NMR spectra were processed using NMRPipe[46]. For NMR analysis, an integrated package of NMR tools named MagRO-NMRViewJ, version 2.01.41[47], on NMRView was used[48].

For the competition experiments, $^1$H-$^{15}$N HSQC spectra were measured at 293 K for 60 μM [$^{15}$N]-hPHD in the presence of hDPPA3$_{81-118}$ and/or the H3$_{1-37W}$ peptide at molar ratios (hPHD:hPDDA3:H3) of 1:0:0, 1:2:0, 1:0:2, 1:2:2 and 1:2:8.

## ITC measurements

Microcal PEAQ-ITC (Malvern) was used for ITC measurements. Wild-type and mutants of hPHD and hDPPA3 were dissolved in ITC buffer (10 mM HEPES [pH 7.5] buffer containing 150 mM NaCl and 0.25 mM TCEP). All measurements were carried out at 293 K. The data were analyzed with Microcal PEAQ-ITC analysis software using a one-site model. For each interaction, at least three independent titration experiments were performed to show the dissociation constants with mean standard deviation.

## CD

Far-UV circular dichroism (CD) spectra were obtained using a JASCO J-1100 model spectrometer. All samples were prepared at a concentration of 20 μM, dissolved in 10 mM HEPES [pH7.5] buffer containing 150 mM

NaCl, 0.25 mM TCEP. The measurements were performed at 293 K with a path length of 1 mm.

## SEC-SAXS

SAXS data were collected on Photon Factory BL-10C using an HPLC Nexera/Prominence-I (Shimazu) integrated SAXS set-up[49]. 50 μl of 12 mg/ml hPHD and hPHD:hDPPA3$_{81-118}$ complex and 20 mg/ml hDPPA3$_{81-118}$ were loaded onto a Superdex® 200 Increase 5/150 GL (Cytiva) pre-equilibrated with 20 mM Tris-HCl (pH 7.5), 150 mM NaCl, 2 mM DTT, 10 μM zinc acetate and 5% glycerol at a flow rate of 0.25 ml/min at 20 °C. The flow rate was reduced to 0.025 ml/min at an elution volume of 1.9–2.8 ml. X-ray scattering data were collected every 20 s on a PILATUS3 2 M detector over an angular range of $q_{min} = 0.00690$ Å$^{-1}$ to $q_{max} = 0.27815$ Å$^{-1}$. The UV spectra at the range of 200–450 nm were recorded every 10 s. Circular averaging and buffer subtraction were carried out using the program SAngler[50] to obtain one-dimensional scattering data $I(q)$ as a function of $q$ ($q = 4\pi\sin\theta/\lambda$, where $2\theta$ is the scattering angle and $\lambda$ is the X-ray wavelength 1.5 Å). The scattering intensity was normalized on an absolute scale using the scattering intensity of water[41]. The multiple concentrations of the scattering data around the peak at A280, namely the ascending and descending parts of the chromatography peak, and $I(0)$ were extrapolated to zero-concentration using MOLASS[51]. The molecular mass of the measured proteins was estimated using the empirical volume of correlation, $V_c$, showing no aggregation of the measured sample[34]. The radius of gyration $R_g$ and forward scattering intensity $I(0)$ were estimated from the Guinier plot of $I(q)$ in the smaller-angle region of $qR_g < 1.3$. The distance distribution function, $P(r)$, was calculated using the program GNOM[52]. The maximum particle dimension $D_{max}$ was estimated from the $P(r)$ function as the distance r for which $P(r) = 0$. The scattering profile of the crystal structure of hPHD:hDPPA3$_{81-118}$ was computed using CRYSOL[53] software. Ab initio model of hPHD:hDPPA3$_{81-118}$ was created using GASBOR and DAMAVER[54,55].

## In vitro ubiquitination assay

Protein expression in E. coli and purification of mouse UBA1 (E1), human UBE2D3 (E2), human UHRF1 (E3), C-terminal FLAG tagged-H3$_{1-37W}$ and ubiquitin were performed according to previous reports[17]. The ubiquitination reaction mixtures contained 100 μM ubiquitin, 200 nM E1, 8 μM E2, 3 μM E3, 5 mM ATP, and 20 μM C-terminal FLAG tagged-H3$_{1-37W}$ in the presence and absence of hDPPA3$_{81-118}$ in ubiquitination reaction buffer (50 mM Tris-HCl [pH 8.0], 50 mM NaCl, 5 mM MgCl$_2$, 0.1% Triton X-100, 2 mM DTT). The mixture was incubated at 30 °C for 3 h and the reaction was stopped by adding 3 × SDS loading buffer. The reaction was analyzed by SDS-PAGE, followed by Western blotting using a 1/5000 diluted anti-FLAG antibody (Cell Signaling Technology, #2368).

## Xenopus interphase egg extracts and purification of chromatin

Xenopus laevis was purchased from Kato-S Kagaku and handled according to animal care regulations at the University of Tokyo. Interphase egg extracts were prepared as described previously[12]. Unfertilized Xenopus eggs were dejellied in 2.5% thioglycolic acid-NaOH (pH 8.2) and washed three times in 0.2 × MMR buffer (5 mM HEPES-KOH [pH 7.6], 100 mM NaCl, 2 mM KCl, 0.1 mM EDTA, 1 mM MgCl$_2$, 2 mM CaCl$_2$). After activation in 1 × MMR supplemented with 0.3 μg/ml calcium ionophore, eggs were washed four times with EB buffer (10 mM HEPES-KOH [pH 7.7], 100 mM KCl, 0.1 mM CaCl$_2$, 1 mM MgCl$_2$, 50 mM sucrose). Packed eggs were crushed by centrifugation (BECKMAN, Avanti J-E, JS13.1 swinging rotor) for 20 min at 18,973 × g. Egg extracts were supplemented with 50 μg/ml cycloheximide, 20 μg/ml cytochalasin B, 1 mM DTT, 2 μg/ml aprotinin, and 50 μg/ml leupeptin, and clarified for 20 min at 48,400 × g (Hitachi, CP100NX, P55ST2 swinging rotor). The cytoplasmic extracts were aliquoted and stored at −80 °C. Chromatin purification after incubation in egg extracts was performed as previously described with modifications. Sperm nuclei were incubated in egg extracts supplemented with an ATP regeneration system (20 mM phosphocreatine, 4 mM ATP, and 5 μg/ml creatine

phosphokinase) at 3000–4000 nuclei/μl at 22 °C. Aliquots (15 μl) were diluted with 150–200 μl of chromatin purification buffer (CPB; 50 mM KCl, 5 mM MgCl₂, 20 mM HEPES-KOH [pH 7.7]) containing 0.1% NP-40, 2% sucrose, and 2 mM NEM. After incubation on ice for 5 min, the extracts were layered over 1.5 ml CPB containing 30% sucrose and centrifuged at $15,000 \times g$ for 10 min at 4 °C. Chromatin pellets were resuspended in 1 × Laemmli sample buffer, heated for 5 min and analyzed by SDS-PAGE. Recombinant FLAG-tagged mDPPA3 and hDPPA3 were added to egg extracts at 0.5–1.0 μM. For FLAG-tagged protein expression in insect cells, 3×FLAG-tagged *mDppa3* or *hDppa3* were sub-cloned into pVL1392 vector. Baculoviruses were produced using a BD BaculoGold Transfection Kit and a BestBac Transfection Kit (BD Biosciences) following the manufacturer's protocol. Proteins were expressed in Sf9 insect cells by infection with viruses expressing 3×FLAG-tagged mDPPA3 WT or its mutants for 72 h at 27 °C. Sf9 cells from a 750 ml culture were collected and lysed by resuspending them in 30 ml lysis buffer (20 mM Tris-HCl [pH 8.0], 100 mM KCl, 5 mM MgCl₂, 10% glycerol, 1% NP-40, 1 mM DTT, 5 μg/ml leupeptin, 2 μg/ml aprotinin, 20 μg/ml trypsin inhibitor, 100 μg/ml phenylmethylsulfonyl fluoride [PMSF]), followed by incubation on ice for 10 min. The soluble fraction was obtained after centrifugation of the lysate at $15,000 \times g$ for 15 min at 4 °C. The soluble fraction was incubated with 250 μl anti-FLAG M2 affinity resin equilibrated with lysis buffer for 4 h at 4 °C. The beads were collected and washed with 10 ml wash buffer and then with 5 ml of EB [20 mM HEPES-KOH (pH 7.5), 100 mM KCl, 5 mM MgCl₂] containing 1 mM DTT. Each recombinant protein was eluted twice in 250 μl EB containing 1 mM DTT and 250 μg/ml 3 × FLAG peptide (Sigma-Aldrich). The eluates were pooled and concentrated using a Vivaspin 500 (GE Healthcare).

## Statistics and reproducibility

All biochemical and biophysical experiments were repeated at least three times.

## Data availability

Coordinate of atomic model of human UHRF1 PHD finger in complex with human DPPA3 was deposited in the Protein Data Bank with accession code 8WMS. All data needed to evaluate the conclusions in the paper are presented in the paper and/or Supplementary Materials. Additional data related to this paper may be requested from the authors.

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

## Acknowledgements

We would like to thank the beamline staff at the Photon Factory for X-ray data and SAXS collections. This study was supported by MEXT/JSPS KAKENHI (18H02392, 19H05294, 19H05741, and 24K01967 to K.A., 19H05285 and 21H00272 to A.N., and 23K05720 to T.K.), PRESTO (14530337) from JST to K.A., a grant for 2021–2023 Strategic Research Promotion (No. SK201904) of Yokohama City University to K.A., and Research Development Fund of Yokohama City University to T.K.

## Author contributions

K.A. conceived the study and experimental design, analyzed the experiments, and wrote the manuscript. N.S. and K.A. performed the protein purification, X-ray crystallography, SAXS measurements, and ITC experiments. N.S. performed CD experiments. N.S. and N.N. performed in vitro ubiquitination assays. S.H. and T.K. performed NMR experiments and analyzed the data. Y.C., M.H.I., M.N., and A.N. performed the biochemical assays using *Xenopus* egg extracts.

## Competing interests

The authors declare no competing interests.
