## [Peer Review File · Communications Biology]

Reviewers' comments:

Reviewer #1 (Remarks to the Author):

In this study, Shiraishi et al. have characterized the binding of UHRF1 and DPPA3 in human. This study shows a remarkable difference between the mouse and human counterparts of DPPA3 in its ability to bind to UHRF1 PHD finger. In addition, it also shows that a single helical conformation in human DPPA3 (res. 88-107) is responsible for the reduced affinity for UHRF1. The experiments are well planned and executed, and the manuscript is nicely written.

I have just one suggestion. It appears that the major difference in the helical region of human and mouse DPPA3 is due to the substitution of proline with lysine at the 95th position in human. A similar substitution (with a positively charged residue) is also present in *Bos taurus*, *Gorilla gorilla*, *Saimiri boliviensis*, *Puma concolor*, *Nomascus leucogenys*, *Crocota crocuta*, *Physeter macrocephalus*, and *Acinonyx jubatus*, whereas it is conserved in *Mus musculus*, *Rattus norvegicus*, and *Cricetulus griseus*. I would suggest the authors to include K95P mutant of human DPPA3 in this study to see if this lysine to proline substitution is able to dislodge UHRF1 from chromatin.

Minor suggestion: Figures should be as per citation order. E.g., The citation "Figure 1" should come before "Figure 2".

Reviewer #2 (Remarks to the Author):

In the manuscript COMMSBIO-23-4612 "Structure of human DPPA3 bound to the UHRF1 PHD finger reveals its functional and structural differences from mouse DPPA3" the authors present a crystal structure of a fragment of human DPPA3 (also called Stella/PGC7) in complex with the PHD domain of human UHRF1, a ubiquitin E3 ligase that is essential for DNA methylation by mediating the recruitment of the maintenance DNA methyltransferase DNMT1 to chromatin. The authors further present a number of biochemical experiments with inactivating mutants in the DPPA3 fragment to test the predictions from the structure. Several recent studies found the mouse DPPA3 protein to be an inhibitor of UHRF1 and its function in DNA methylation, both by binding to its PHD domain with high affinity thereby outcompeting and inhibiting the binding of UHRF1 to its ubiquitylation targets histone H3 and PAF15, and by sequestering UHRF1 in the cytosol. In the mouse, DPPA3 is only expressed in oocytes, primordial germ cells, and in pre-implantation embryos - the DPPA3/UHRF1 interaction is therefore a key mechanism that controls low DNA methylation levels in germ cells and during early embryonic development.

A key finding of this manuscript is that the association of the human DPPA3 with the PHD domain of UHRF1 presented here differs substantially from that of the mouse DPPA3 protein, which the same group presented in a previous publication (Hata et al., 2022, *Nucleic Acids Res.* 50: 12527-12542). This difference is mainly due to an induced alpha-helical stretch in DPPA3 that assumes a different

fold in the human and mouse proteins. This results in a substantially weaker affinity of the human DPPA3 to the UHRF1 PHD domain than the mouse DPPA3, resulting in a much weaker inhibitory activity of the human DPPA3 toward UHRF1 functions. The experimental approach and results are all sound, and I find this study very interesting and timely, especially since it unearths the differences between the human and mouse DPPA3. While the structural part is already strong, I think the consequences of these differences for the biological activity of the human and mouse proteins could still be worked out a bit better, to really demonstrate that the human DPPA3 differs in function from the mouse protein. Furthermore, I think that a few more tests that the structure is correct need to be conducted. Please see my comments below.

Major points:

> A major concern regarding the interpretation of the results is the dimer formation of the hPHD:hDPPA3 complex in the crystal, that the authors highlight themselves (lines 167-172). It is possible that the crystal packing forces the human DPPA3 into forming a straight alpha-helix instead of a L-shaped one like in the mouse protein. In addition the N- and C-terminal parts of the DPPA3 alpha-helix interact with the pre- and core-PHD ends of the two PHD fragments. The authors must conduct additional tests to exclude that the straight alpha-helical conformation of the hDPPA3 fragment on the PHD finger is an artefact of the crystal and to test what parts of DPPA3 interact with the PHD domain.

I suggest that the authors design a series of point mutants in the alpha-helical part of the hDPPA3 construct in which they mutate residues pointing towards and away from the pre-PHD and pointing towards the interaction surface between the alpha-helices observed in the crystal. These should then be tested in ITC measurements and other functional assays to observe their effects on PHD binding and UHRF1 activity. In addition it would be interesting to generate and test mutants in the hDPPA3 fragment that correspond to mutations of residues introduced in the alpha-helical parts of the mouse DPPA3 (if these can be identified) that interfere with the binding to the mouse PHD surface (see Figure 4 in publication Hata et al., 2022, *Nucleic Acids Res.* 50: 12527-12542). If the binding mode of the mouse and human DPPA3 is different these should not have an effect in hDPPA3, but other mutations might have.

> The authors should also further test whether the hPHD:hDPPA3 complex indeed forms a dimer in solution. This could be done by generating two different hPHD constructs each carrying a different small affinity tag, and performing co-IPs with these two proteins in the presence and absence of different amounts of the hDPPA3 fragment to test whether addition of the fragment induces an interaction between the differently tagged hPHD domains. If a dimerization is observed, the alpha-helix mutants as above could be tested in this assay.

> another major point that should be addressed with respect to characterising the difference between human and mouse DPPA3 is to test to what extent the human DPPA3 can shuttle human UHRF1 into the cytosol, similar to the experiments conducted for the mouse protein in Figure 6 in Hata et al., 2022, *Nucleic Acids Res.* 50: 12527-12542. The authors can use their mDPPA3 knock out mouse ES cell system for this, but modify it to express human UHRF1-GFP and inducible

mScarlet-fused WT and mutant human DPPA3 proteins. This way they can do a side-by-side comparison of the ability of mouse and human DPPA3 to sequester UHRF1 in the cytosol in the same system. This would further strengthen their manuscript with a relevant *in vivo* readout of the differences in addition to the *in vitro* ubiquitylation, chromatin binding and DNA methylation assays.

Minor points:

- > Based on the prior knowledge, the human and the previously used mouse DPPA3 fragments have slightly different designs (mDPPA3 ranges from aa 76-128 and hDPPA3 ranges from aa 81-118). The constructs are therefore not directly comparable. Especially the N-terminal part corresponding to aa 76-84 in the mouse construct are missing in the human protein fragment. While these are not assuming a clearly folded structure in the mouse protein they are still interacting with the surface of the UHRF1 PHD domain (see Figure 2b in Hata et al., 2022, *Nucleic Acids Res.* 50: 12527-12542) and could therefore contribute to the binding between mDPPA3 and the PHD. The authors should acknowledge this in the text when comparing the affinities of the mouse and the human DPPA3 fragments.
- > For clarity the authors should also include structures of the human UHRF1 PHD domain in complex with the histone H3 and PAF15 N-termini in Figure 2 to enable comparison to the different binding modes of hDPPA3 and mDPPA3.
- > Figure 4b, lower panel: the authors should note the concentration of the hDPPA3 WT and mutants used in this experiment in the figure legend.
- > Supplementary Figure 4b: the difference between hDPPA3 and mDPPA3 is not very striking, the authors should repeat this experiment with titrations for both hDPPA3 and mDPPA3 (at 20, 40, 100 μ M as in Figure 4b) to clearly work out the difference.
- > Figure 5b: it seems that in this panel the right two lanes are mislabelled as '0.5 μ M hDPPA3' since in the text the authors refer to this figure as '1.0 μ M hDPPA3'. If this experiment was really conducted with 0.5 μ M hDPPA3 then it should be repeated with 1.0 μ M hDPPA3.
- > The authors should check their manuscript for some remaining small typos or misspelling errors.

Point-by-point responses to reviewer comments.

I thank the two reviewers, who took valuable time to evaluate our paper.

I have addressed all the criticisms of the reviewers. Please see our point-by-point responses below.

Reviewer #1 (Remarks to the Author):

In this study, Shiraishi et al. have characterized the binding of UHRF1 and DPPA3 in human. This study shows a remarkable difference between the mouse and human counterparts of DPPA3 in its ability to bind to UHRF1 PHD finger. In addition, it also shows that a single helical conformation in human DPPA3 (res. 88-107) is responsible for the reduced affinity for UHRF1. The experiments are well planned and executed, and the manuscript is nicely written.

*I have just one suggestion. It appears that the major difference in the helical region of human and mouse DPPA3 is due to the substitution of proline with lysine at the 95th position in human. A similar substitution (with a positively charged residue) is also present in *Bos taurus*, *Gorilla gorilla*, *Saimiri boliviensis*, *Puma concolor*, *Nomascus leucogenys*, *Crocota crocuta*, *Physeter macrocephalus*, and *Acinonyx jubatus*, whereas it is conserved in *Mus musculus*, *Rattus norvegicus*, and *Cricetulus griseus*. I would suggest the authors to include K95P mutant of human DPPA3 in this study to see if this lysine to proline substitution is able to dislodge UHRF1 from chromatin.*

Thank you for this suggestion. As the reviewer pointed out, we also considered that the Pro residue in mice plays a key role in the formation of two helices and has a high binding affinity to UHRF1 PHD. To test this hypothesis, we introduced the K95P mutation in human DPPA3 (changed to the mouse type) and performed ITC experiments (right figure). However, this mutation did not enhance the binding affinity of hDPPA3 to hPHD. According to the AF2

prediction of hDPPA3 containing the K95P mutation, only one predicted structure of the mutant exhibited a mouse-type two-helix structure (see bottom figure). These data indicated that the difference in the helical structural composition of human and mouse DPPA3 is driven by a more complicated mechanism than expected. We have added the analysis of the K95P mutant of hDPPA3 in Supplementary Figure 7 and the Discussion section, lines 340-348.

Minor suggestion: Figures should be as per citation order. E.g., The citation “Figure 1” should come before “Figure 2”.

Thank you for pointing this out. In the revised manuscript, we have changed the previous Figures 1a and 1b to 1b and 1a, respectively.

Reviewer #2 (Remarks to the Author):

In the manuscript COMMSBIO-23-4612 "Structure of human DPPA3 bound to the UHRF1 PHD finger reveals its functional and structural differences from mouse DPPA3" the authors present a crystal structure of a fragment of human DPPA3 (also called Stella/PGC7) in complex with the PHD domain of human UHRF1, a ubiquitin E3 ligase that is essential for DNA methylation by mediating the recruitment of the maintenance DNA methyltransferase DNMT1 to chromatin. The authors further present a number of biochemical experiments with inactivating mutants in the DPPA3 fragment to test the predictions from the structure. Several recent studies found the mouse DPPA3 protein to be an inhibitor of UHRF1 and its function in DNA methylation, both by binding to its PHD domain with high affinity thereby outcompeting and inhibiting the binding of UHRF1 to its ubiquitylation targets histone H3 and PAF15, and by sequestering UHRF1 in the cytosol. In the mouse, DPPA3 is only expressed in oocytes, primordial germ cells, and in pre-implantation embryos - the DPPA3/UHRF1 interaction is therefore a key mechanism that controls low DNA methylation levels in germ cells and during early embryonic development.

A key finding of this manuscript is that the association of the human DPPA3 with the PHD domain of UHRF1 presented here differs substantially from that of the mouse DPPA3 protein, which the same group presented in a previous publication (Hata et al., 2022, Nucleic Acids Res. 50: 12527-12542). This difference is mainly due to an induced alpha-helical stretch in DPPA3 that assumes a different fold in the human and mouse proteins. This results in a substantially weaker affinity of the human DPPA3 to the UHRF1 PHD domain than the mouse DPPA3, resulting in a much weaker inhibitory activity of the human DPPA3 toward UHRF1 functions. The experimental approach and results are all sound, and I find this study very interesting and timely, especially since it unearths the differences between the human and mouse DPPA3. While the structural part

is already strong, I think the consequences of these differences for the biological activity of the human and mouse proteins could still be worked out a bit better, to really demonstrate that the human DPPA3 differs in function from the mouse protein. Furthermore, I think that a few more tests that the structure is correct need to be conducted. Please see my comments below.

Major points:

> A major concern regarding the interpretation of the results is the dimer formation of the hPHD:hDPPA3 complex in the crystal, that the authors highlight themselves (lines 167-172). It is possible that the crystal packing forces the human DPPA3 into forming a straight alpha-helix instead of a L-shaped one like in the mouse protein. In addition the N- and C-terminal parts of the DPPA3 alpha-helix interact with the pre- and core-PHD ends of the two PHD fragments. The authors must conduct additional tests to exclude that the straight alpha-helical conformation of the hDPPA3 fragment on the PHD finger is an artefact of the crystal and to test what parts of DPPA3 interact with the PHD domain.

I suggest that the authors design a series of point mutants in the alpha-helical part of the hDPPA3 construct in which they mutate residues pointing towards and away from the pre-PHD and pointing towards the interaction surface between the alpha-helices observed in the crystal. These should then be tested in ITC measurements and other functional assays to observe their effects on PHD binding and UHRF1 activity. In addition it would be interesting to generate and test mutants in the hDPPA3 fragment that correspond to mutations of residues introduced in the alpha-helical parts of the mouse DPPA3 (if these can be identified) that interfere with the binding to the mouse PHD surface (see Figure 4 in publication Hata et al., 2022, Nucleic Acids Res. 50: 12527-12542). If the binding mode of the mouse and human DPPA3 is different these should not have an effect in hDPPA3, but other mutations might have.

Thank you for your valuable comments. We designed the following mutations in the α -helix of hDPPA3 to test its interaction with hPHD:

1. R98A/M102A: These amino acid residues are located in the C-terminal region of the α -helix of hDPPA3 and interact with the hPHD finger of a symmetrical molecule in the crystal.
2. M96A/L99A: These residues were positioned at the α -helix dimer interface. This mutant was expected to disrupt dimer formation as observed in the crystal structure.
3. R93P/A97P: These mutations disrupt the α -helical structure of DPPA3. These side chains were exposed to the solvent region and did not interact with the protein moiety.

ITC experiments indicated that mutations 1 (R98A/M102A) and 2 (M96A/L99A) in hDPPA3 did not affect the binding to hPHD (Supplementary Figure 4a). These data indicate that the dimer

formation of hDPPA3:hPHD, as observed in the crystal, is not representative of the native state in solution, which is caused by the crystal packing. Interestingly, mutation 3 (R93P/A97P), which led to the disruption of the α -helix structure of hDPPA3, significantly reduced its binding affinity to hPHD ($K_D = 9.39 \mu\text{M}$, Supplementary Figure 4a). This indicates that the induction of the helical structure of hDPPA3 is important for its binding to hPHD.

We have added the ITC data to Supplementary Figure 4a and the related text in lines 215-224.

> The authors should also further test whether the hPHD:hDPPA3 complex indeed forms a dimer in solution. This could be done by generating two different hPHD constructs each carrying a different small affinity tag, and performing co-IPs with these two proteins in the presence and absence of different amounts of the hDPPA3 fragment to test whether addition of the fragment induces an interaction between the differently tagged hPHD domains. If a dimerization is observed, the alpha-helix mutants as above could be tested in this assay.

We apologize for not clearly explaining the benefits of SAXS experiments in our previous manuscript. We appreciate the opportunity to elaborate on why SAXS was instrumental in our study.

SAXS is one of the most versatile and informative experiments available for analyzing the solution structure, oligomeric state, conformational changes and flexibility of biomacromolecules at a scale ranging from a few Å to hundreds of nm. SAXS is now extensively used in structural biology research. In our study, SAXS played a crucial role in conclusively determining the formation of hPHD:hDPPA3 with a 1:1 stoichiometry. Notably, the dummy atom model of hPHD:hDPPA3 complex in solution is identical to the crystal structure, which strongly supports

our finding that complex formation of the hPHD:hDPPA3 is 1:1 stoichiometry and hDPPA3 adopts a single α -helix conformation upon binding to hPHD. Furthermore, SAXS experiment was performed using non-tagged proteins. This approach significantly excludes the potential for artifacts arising from non-specific interactions between tags and proteins, thus ensuring the authenticity of our findings. Given these points, we believe that our SAXS data comprehensively address the concerns raised and underscores the robustness of our experimental design. We hope that this explanation clarifies the reviewer's concern.

We have added a brief explanation of SAXS in lines 174-176.

> another major point that should be addressed with respect to characterising the difference between human and mouse DPPA3 is to test to what extent the human DPPA3 can shuttle human UHRF1 into the cytosol, similar to the experiments conducted for the mouse protein in Figure 6 in Hata et al., 2022, Nucleic Acids Res. 50: 12527-12542. The authors can use their mDPPA3 knock out mouse ES cell system for this, but modify it to express human UHRF1-GFP and inducible mScarlet-fused WT and mutant human DPPA3 proteins. This way they can do a side-by-side comparison of the ability of mouse and human DPPA3 to sequester UHRF1 in the cytosol in the same system. This would further strengthen their manuscript with a relevant in vivo readout of the differences in addition to the in vitro ubiquitylation, chromatin binding and DNA methylation assays.

Thank you for your comment. The referee is correct that, ideally, one would analyze the effect of hDPPA3 on the localization of UHRF1 in mammalian cells. However, the experiment proposed by the reviewer requires the introduction of hUHRF1 and hDPPA3 into mESCs at appropriate expression levels, which requires several optimizations. We feel this would be outside the scope of the current study.

We are exploring the possibility that the liquid-liquid phase separation of human DPPA3 regulates the function of UHRF1. To further investigate this, we plan to perform experiments in the future. Notably, sequence analysis using FuzDrop (<https://fuzdrop.bio.unipd.it>) suggested that human DPPA3 has a higher potential for liquid-liquid phase separation compared to mouse DPPA3, as shown in the lower figure.

We have added this perspective to the Discussion section, lines 304-310, of the revised manuscript.

Prediction of droplet formation of human (left) and mouse (right) DPPA3 analyzed by FuzDrop (<https://fuzdrop.bio.unipd.it>)

Minor points:

> *Based on the prior knowledge, the human and the previously used mouse DPPA3 fragments have slightly different designs (mDPPA3 ranges from aa 76-128 and hDPPA3 ranges from aa 81-118). The constructs are therefore not directly comparable. Especially the N-terminal part corresponding to aa 76-84 in the mouse construct are missing in the human protein fragment. While these are not assuming a clearly folded structure in the mouse protein they are still interacting with the surface of the UHRF1 PHD domain (see Figure 2b in Hata et al., 2022, Nucleic Acids Res. 50: 12527-12542) and could therefore contribute to the binding between mDPPA3 and the PHD. The authors should acknowledge this in the text when comparing the affinities of the mouse and the human DPPA3 fragments.*

Thank you for the comment. We previously identified the minimum region of mDPPA3 required for binding to mPHD using NMR solution structure analysis; aa76-84 was not structurally converged, indicating that this region is not essential for binding to UHRF1. Consequently, we focused on residues 81-118 of hDPPA3 in this study.

> *For clarity the authors should also include structures of the human UHRF1 PHD domain in complex with the histone H3 and PAF15 N-termini in Figure 2 to enable comparison to the different binding modes of hDPPA3 and mDPPA3.*

We have added the structures of hPHD:H3 and hPHD:PAF15 to Figure 2B.

> *Figure 4b, lower panel: the authors should note the concentration of the hDPPA3 WT and mutants used in this experiment in the figure legend.*

Thank you for pointing it out. We have amended the figure legend.

> *Supplementary Figure 4b: the difference between hDPPA3 and mDPPA3 is not very striking, the authors should repeat this experiment with titrations for both hDPPA3 and mDPPA3 (at 20, 40, 100 μ M as in Figure 4b) to clearly work out the difference.*

Thank you for this suggestion. We have conducted the experiment and the data have been added to Figure 4B. We are convinced that the Western blot data clearly demonstrate that the inhibitory effect of mDPPA3 is significantly stronger than that of hDPPA3.

> *Figure 5b: it seems that in this panel the right two lanes are mislabelled as '0.5 μ M hDPPA3' since in the text the authors refer to this figure as '1.0 μ M hDPPA3'. If this experiment was really conducted with 0.5 μ M hDPPA3 then it should be repeated with 1.0 μ M hDPPA3.*

We apologize for this mislabeling. We have amended the label in the Figure 5b.

> *The authors should check their manuscript for some remaining small typos or misspelling*

errors.

We apologize for these errors. We have carefully checked these errors.

REVIEWERS' COMMENTS:

Reviewer #1 (Remarks to the Author):

The authors have addressed all the comments. However, I have a minor comment to make. There is a discrepancy in the legend of figure 1 a,b. Authors should correct that. I recommend this manuscript for publication.

Reviewer #2 (Remarks to the Author):

In the revised version of manuscript COMMSBIO-23-4612A the authors have addressed all my points adequately. It is a very clear and coherent piece of work that is logical in itself and reads very well. I accept that the experiment to introduce human UHRF1 and human DPPA3 into the mESC system to test the effect of wt and mutant hDPPA3 on hUHRF1 localization is beyond the scope of this manuscript.

However, I would suggest to integrate the graphical depiction of the positions of the newly added mutations in the hDPPA3 alpha-helix provided in the rebuttal letter also into Supplementary Figure 4 as this would be helpful for the reader to explain the rationale for the mutations. I have also spotted a few minor errors that need correction. Apart from these points I have no further comments, and once these are fixed I recommend the manuscript for publication.

Minor corrections:

1. Since the order of panels (a) and (b) in Figure 1 were exchanged, the descriptions for (a) and (b) in the figure legend also need to be exchanged.
2. The AlphaFold2 prediction of the DPPA3 helical fold for *Acinonyx jubatus* in Supplementary Figure 6 is shown twice. This should be corrected.
4. The new AlphaFold2 prediction of the impact of the K95P mutation on the human DPPA3 helical fold presented in Supplementary Figure 7b is referenced as Supplementary Figure 8 in the discussion. This should be corrected.

I would like to thank the two reviewers who took valuable time to evaluate our paper and support the publication.

Reviewer #1 (Remarks to the Author):

The authors have addressed all the comments. However, I have a minor comment to make. There is a discrepancy in the legend of figure 1 a,b. Authors should correct that. I recommend this manuscript for publication.

Thank you for pointing this out. We have amended the discrepancy in the figure legends.

Reviewer #2 (Remarks to the Author):

In the revised version of manuscript COMMSBIO-23-4612A the authors have addressed all my points adequately. It is a very clear and coherent piece of work that is logical in itself and reads very well. I accept that the experiment to introduce human UHRF1 and human DPPA3 into the mESC system to test the effect of wt and mutant hDPPA3 on hUHRF1 localization is beyond the scope of this manuscript.

However, I would suggest to integrate the graphical depiction of the positions of the newly added mutations in the hDPPA3 alpha-helix provided in the rebuttal letter also into Supplementary Figure 4 as this would be helpful for the reader to explain the rationale for the mutations. I have also spotted a few minor errors that need correction. Apart from these points I have no further comments, and once these are fixed I recommend the manuscript for publication.

Thank you for this suggestion. We have added the figure in Supplementary Figure 4a.

Minor corrections:

1. Since the order of panels (a) and (b) in Figure 1 were exchanged, the descriptions for (a) and (b) in the figure legend also need to be exchanged.

Thank you for pointing it out. We have amended the figure legends.

2. The AlphaFold2 prediction of the DPPA3 helical fold for Acinonyx jubatus in Supplementary Figure 6 is shown twice. This should be corrected.

Thank you very much for pointing out this error, the duplication of the structure of *Acinonyx jubatus*. We have corrected this duplication.

4. The new AlphaFold2 prediction of the impact of the K95P mutation on the human DPPA3 helical fold presented in Supplementary Figure 7b is referenced as Supplementary Figure 8 in the discussion. This should be corrected.

Thank you for pointing it out. We have amended the figure number accordingly.